# Implementing a choice of pulmonary rehabilitation models in chronic obstructive pulmonary disease (HomeBase2 trial): protocol for a cluster randomised controlled trial

Anne E Holland [ID],[1,2,3] Arwel W Jones [ID],[1] Ajay Mahal,[4] Natasha A Lannin [ID],[5,6] Narelle Cox [ID],[1,3] Graham Hepworth,[7] Paul O'Halloran,[8] Christine F McDonald [ID] [9,10]

For numbered affiliations see end of article.

**Correspondence to**
Dr Anne E Holland;
a.holland@alfred.org.au

## ABSTRACT

**Introduction** There is compelling evidence that either centre-based or home-based pulmonary rehabilitation improves clinical outcomes in chronic obstructive pulmonary disease (COPD). There are known health service and personal barriers which prevent potentially eligible patients from accessing the benefits of pulmonary rehabilitation. The aim of this hybrid effectiveness-implementation trial is to examine the effects of offering patients a choice of pulmonary rehabilitation locations (home or centre) compared with offering only the traditional centre-based model.

**Method and analysis** This is a two-arm cluster randomised, controlled, assessor-blinded trial of 14 centre-based pulmonary rehabilitation services allocated to intervention (offering choice of home-based or centre-based pulmonary rehabilitation) or control (continuing to offer centre-based pulmonary rehabilitation only), stratified by centre-based programme setting (hospital vs non-hospital). 490 participants with COPD will be recruited. Centre-based pulmonary rehabilitation will be delivered according to best practice guidelines including supervised exercise training for 8 weeks. At intervention sites, the home-based pulmonary rehabilitation will be delivered according to an established 8-week model, comprising of one home visit, unsupervised exercise training and telephone calls that build motivation for exercise participation and facilitate self-management. The primary outcome is all-cause, unplanned hospitalisations in the 12 months following rehabilitation. Secondary outcomes include programme completion rates and measurements of 6-minute walk distance, chronic respiratory questionnaire, EQ-5D-5L, dyspnoea-12, physical activity and sedentary time at the end of rehabilitation and 12 months following rehabilitation. Direct healthcare costs, indirect costs and changes in EQ-5D-5L will be used to evaluate cost-effectiveness. A process evaluation will be undertaken to understand how the choice model is implemented and explore sustainability beyond the clinical trial.

**Ethics and dissemination** Alfred Hospital Ethics Committee has approved this protocol. The trial findings

## Strengths and limitations of this study

► This hybrid effectiveness-implementation trial, conducted across 14 sites, will be the first to compare offering a choice of home-based or centre-based pulmonary rehabilitation to centre-based pulmonary rehabilitation only.
► The trial will only include participants with a diagnosis of chronic obstructive pulmonary disease (COPD).
► All participants, including those in the control group, will have the opportunity to achieve clinical benefits from an evidence-based pulmonary rehabilitation programme.
► It is possible that some individuals who are unable or unwilling to attend a centre-based programme may choose not to participate in the trial, as the choice of programme location is only offered after consent is given. Trial participants may therefore not be representative of all those who might benefit from pulmonary rehabilitation.
► The primary outcome is unplanned hospitalisation, the most valued outcome of patients with COPD.

will be published in peer-reviewed journals, submitted for presentation at conferences and disseminated to patients across Australia with support from national lung charities and societies.

**Trial registration number** NCT04217330.

## INTRODUCTION

Chronic obstructive pulmonary disease (COPD) is characterised by persistent respiratory symptoms and airflow limitation.[1] Activity-related dyspnoea and fatigue in COPD ultimately lead to reductions in physical activity and conditioning, which are associated with poor quality of life and exacerbations of respiratory symptoms.[2 3] Acute exacerbations are a leading cause of preventable

BMJ

hospitalisations and account for more than 50% of the costs associated with the treatment of COPD.[4–6] The goals of COPD management are to reduce exacerbations and hospitalisations by improving symptoms, optimising self-care and minimising disease progression.[1]

Pulmonary rehabilitation is defined as 'a comprehensive intervention based on a thorough patient assessment followed by patient-tailored therapies that include, but are not limited to, exercise training, education and behaviour change, designed to improve the physical and psychological condition of people with chronic respiratory disease and to promote the long-term adherence to health-enhancing behaviours'.[7] Programmes have traditionally consisted of twice-weekly attendance as an outpatient in a group setting for 8–12 weeks.[8 9] Clinical trials show that outpatient pulmonary rehabilitation for COPD is highly effective, with level 1 evidence for improved exercise capacity, reduced breathlessness and improved quality of life, regardless of disease severity.[10] Pulmonary rehabilitation also reduces acute exacerbations, decreases hospital days and reduces hospital admissions, with a concomitant reduction in healthcare costs.[11–13]

Despite compelling evidence for the benefits of pulmonary rehabilitation and unequivocal recommendations in COPD guidelines,[8] patient access to pulmonary rehabilitation remains limited worldwide.[14–19] Barriers preventing patients from accessing the benefits of pulmonary rehabilitation occur at a system level (shortage of programmes) and individual level (poor physical mobility, distressing symptoms and inability to travel).[20 21] Between 8% and 50% of patients referred to a pulmonary rehabilitation programme never attend, and of those who do start, between 10% and 32% do not complete a programme.[20] A 2015 American Thoracic Society (ATS)/European Respiratory Society policy statement called on the research community to increase access to pulmonary rehabilitation by establishing the efficacy of alternative models.[21]

A low-cost model of delivering pulmonary rehabilitation directly into the home was developed (HomeBase) in Australia to improve access and uptake.[22] A previous Phase II randomised controlled equivalence trial[23] showed that HomeBase delivered equivalent clinical outcomes (exercise capacity, symptoms and quality of life) to traditional centre-based rehabilitation in COPD with similar costs and better completion rates (91% vs 49%). The clinical implementation of home-based pulmonary rehabilitation programmes has the potential to improve health outcomes, reduce healthcare utilisation and minimise societal costs but should not be considered a replacement for traditional centre-based programmes. Semi-structured interviews with participants who received home-based rehabilitation in the Phase II trial revealed that participants valued the convenience and flexibility of exercise training at home, and the reduced travel burden of a home-based programme.[24] However, 18% of individuals assessed for eligibility to that trial declined to participate because they wanted to be certain they would receive their preferred choice of a traditional centre-based programme.[23]

A recent ATS workshop report identified that the emergence of alternative models of pulmonary rehabilitation pose many unanswered questions for clinical practice, which will be best addressed by prospective clinical implementation trials.[8] At a health service or programme level it is likely that the best outcomes and optimal completion rates will be achieved by offering patients a choice of pulmonary rehabilitation locations. Our previous equivalence trial of home versus centre-based rehabilitation showed that patients who completed pulmonary rehabilitation, regardless of group allocation, were 56% less likely to be hospitalised in the following 12 months and had a longer time to their next hospital admission.[23] Offering the choice of home or centre-based pulmonary rehabilitation may increase programme completion and therefore result in fewer hospitalisations and more patients with COPD achieving improvements in exercise capacity, breathlessness and quality of life. Until now, the offer of choice of pulmonary rehabilitation location versus traditional centre-based only has not been investigated in a clinical trial.

The aim of this hybrid effectiveness-implementation trial is to examine the effects of offering patients a choice of two pulmonary rehabilitation locations (home or centre) compared with offering only the traditional centre-based model. In order to achieve this aim, the objectives of the trial are threefold:

1. To compare the clinical outcomes, acceptance and completion of pulmonary rehabilitation under the two service models (patient choice between centre-based and home-based, and traditional centre-based only);
2. To estimate the costs and compare the cost-effectiveness of the offer of choice of pulmonary rehabilitation location; and
3. To understand how the choice model is implemented and explore sustainability beyond the clinical trial.

## METHODS AND ANALYSIS
### Design
A two-arm cluster randomised, controlled, assessor-blinded trial will be conducted with 14 centre-based pulmonary rehabilitation services across Australia. The overall trial design is depicted in figure 1. This trial protocol is reported in accordance with Standard Protocol Items: Recommendations for Interventional Trials guidelines.[25] The trial has been prospectively registered at ClinicalTrials.gov, will be conducted in accordance with the principles of Good Clinical Practice and will be reported according to Consolidated Standards of Reporting Trials[26] and Standards for reporting implementation studies (StaRI)[27] standards for cluster randomised controlled trials and implementation studies.

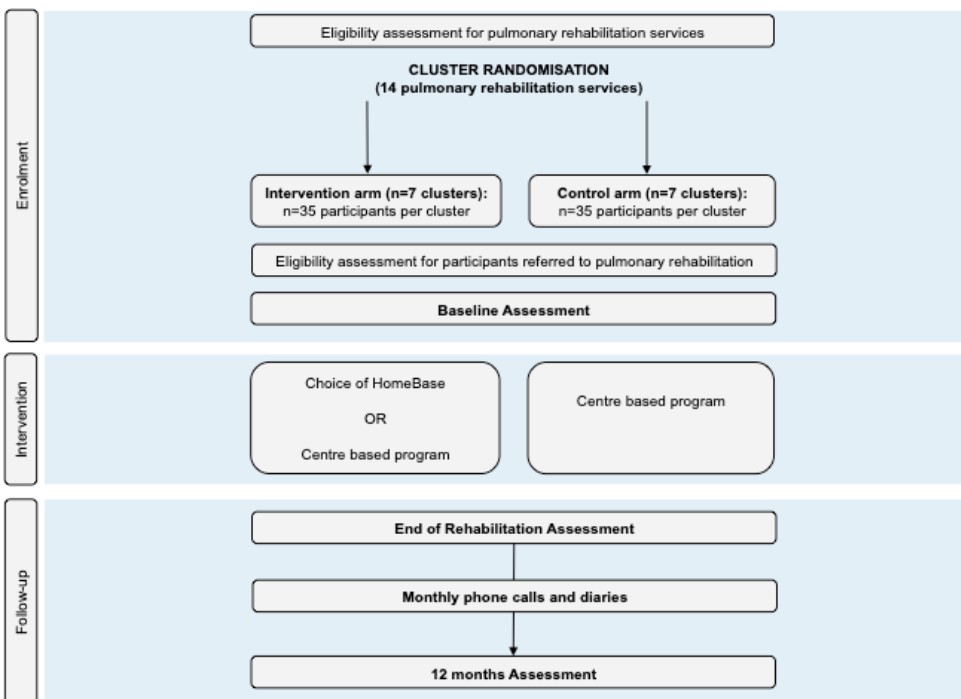

**Figure 1** Design of the HomeBase2 trial.

### Clusters (pulmonary rehabilitation programmes)

Included pulmonary rehabilitation services must provide outpatient programmes that admit at least 50 people with COPD each year and must be able to demonstrate established protocols for centre-based pulmonary rehabilitation according to national guidelines.[28] These sites are representative of the facilities where pulmonary rehabilitation is delivered in Australia, including community programmes, and large and medium sized hospitals.

### Randomisation

Eligible pulmonary rehabilitation services will be randomised 1:1 to either the intervention group (offering choice of home-based or centre-based pulmonary rehabilitation) or control group (continuing to offer only centre-based pulmonary rehabilitation) stratified by setting of existing centre-based pulmonary rehabilitation (hospital vs non-hospital). Such strata have been chosen to balance for potential factors related to implementation of choice of rehabilitation models (eg, local staff capacity and infrastructure) or differences in sociodemographic profiles in areas where hospital or non-hospital programmes are usually situated (eg, urban vs rural)[29] and most common reasons affecting attendance at centre-based rehabilitation or uptake of home-based rehabilitation (eg, accessibility of/distance to hospital venues).[20 24] Randomisation (sequence generation and allocation concealment) will be performed using a secure online randomisation service managed by an independent agency. Block randomisation is being used to both balance cluster allocation within strata and to help ensure the methods (in particular the block size) can remain consistent throughout the trial (eg, should clusters withdraw from the trial).

### Blinding

Research staff: Site principal investigators at each cluster will be notified of their allocation by the research team. Clinicians within sites who are involved in delivery of the intervention will not be blinded to group allocation. However, research staff within clusters that conduct post-intervention assessments and enter questionnaire data into the trial database will be blinded to group allocation.

Participants: Participants will be blinded to group allocation. Patients will be informed that this study is examining the impact of pulmonary rehabilitation models on patient and health system outcomes. The specific nature of the models of interest, or the intervention group to which their site has been allocated will not be disclosed. As this is a study of behaviour (the impact of offering choice of pulmonary rehabilitation models), limited disclosure of the study aim regarding comparison of models is necessary. To truly understand and measure the impact of offering choice of programme models, the offer of choice in the intervention groups will be made after participant consent for the trial has been obtained, so that the choice can be recorded.

### Participants

To be eligible for inclusion at the 14 pulmonary rehabilitation services participants must (i) have a diagnosis of COPD confirmed on spirometry; (ii) be able to read, write and speak English; (iii) be able to provide informed consent. Exclusion criteria include: (i) attendance at pulmonary rehabilitation in the last 12 months; (ii) comorbidities which preclude exercise training (with reference to absolute and relative contraindications for

field walking tests)[30]; (iii) inability to provide informed consent.

## Recruitment

Eligible participants will be identified through referrals to pulmonary rehabilitation programmes at participating clusters. To ensure that referrals are maximised across all sites, respiratory and general physicians at their affiliated hospitals will receive 3 monthly reminders to refer patients with COPD to pulmonary rehabilitation, consistent with current guidelines for best practice care (COPD-X).[31] Screening will be conducted by the site investigator, who is a staff member in the pulmonary rehabilitation programme. Potential participants will be sent an information flyer about the study prior to their initial pulmonary rehabilitation appointment. They will be invited to discuss the study further at their initial appointment, or by telephone.

## Delivery of pulmonary rehabilitation

Participants in both groups will receive a copy of the 'Better Living with Exercise' booklet, produced by Lung Foundation Australia. The 'Better Living with Exercise' booklet assists people with COPD to work with their health professionals to optimise their exercise programme, based on their individual health conditions, goals and interests. All participants will also receive details to access an online version of the 'Better Living with COPD; A Patient Guide' on the Lung Foundation Australia website. The 'Better Living with COPD: A Patient Guide' (Third Edition)[32] was developed to support patients with COPD to better understand and manage their condition. At the completion of the rehabilitation period participants will follow usual care at the trial site which may or may not include a referral to a locally available supervised exercise maintenance programme to promote ongoing exercise adherence. The number of participants who opt to join maintenance programmes will be recorded.

## Control group

Pulmonary rehabilitation programmes assigned to the control group will offer eligible participants the opportunity to participate in their usual centre-based pulmonary rehabilitation, delivered according to current best practice guidelines.[28] This will be an 8-week, twice weekly outpatient group-based supervised programme, with individually prescribed exercise training and self-management education. At least 30 min of aerobic exercise training will be performed each session, plus resistance exercises using functional activities such as stair climbing and sit-to-stand practice, as well as free weights for the upper limbs. Participants will also be encouraged to exercise at home on three occasions each week and to record this in the Lung Foundation Australia 'Better Living with Exercise' resource diary. Programmes may also include self-management training in structured (lecture-based) and unstructured disease management education on a group

and/or 1:1 basis, in accordance with existing clinical practice at the trial site.

## Intervention group

Pulmonary rehabilitation programmes assigned to the intervention group will offer eligible participants the choice of participating in an 8-week programme of either home-based or traditional centre-based pulmonary rehabilitation (see control group). Home-based rehabilitation will be delivered according to our protocol (HomeBase), which has been published in detail previously.[22 23 33] Briefly, HomeBase includes the following components:

*Home visit*: The programme commences with one home visit by a physiotherapist experienced in pulmonary rehabilitation during which exercise goals are established, the first exercise session is supervised and aspects of the self-management programme that cannot be adequately dealt with during subsequent telephone calls (eg, review of inhaler technique), are covered.

*Exercise training*: Participants will follow an aerobic and strength training programme, which will be supervised for the first session and unsupervised thereafter. For the aerobic training, a walking programme will be prescribed with participants set a walking distance or number of steps to be completed in a given time via a pedometer and on the basis of a baseline 6-minute walk test (initial target intensity of 80% of the speed walked in the test). At least 30 min of aerobic exercise is recommended for each session, for at least five sessions per week. Strength training will use functional activities and equipment that are readily accessible in the home environment (eg, sit-to-stand from a dining chair, water bottles for upper limb weights). Participants in the intervention group who choose home-based pulmonary rehabilitation will not have to return their pedometers at end of the programme and can use the device for the entire study duration.

*Home exercise diary*: The initial goals set for aerobic and resistance exercise and details of the supervised exercise session during the home visit will be recorded in a home exercise diary. Participants will be encouraged to document all subsequent programme goals (exercise or health) and completed unsupervised exercise sessions, including duration and distance walked and the number and type of resistance exercises performed, in their diary each week. We have shown a significant relationship between diary-documented exercise sessions and device measured physical activity.[34] Participants will also be asked to record what barriers may prevent them from achieving their goals or exercise and how they anticipate they may overcome these barriers. Prior to commencing each unsupervised exercise session, the home exercise diary prompts the participant to check whether they have any symptoms of moderate or severe exacerbation and to contact the pulmonary rehabilitation clinician at the site if symptomatic.

*Weekly telephone calls*: Following the home visit in week one, each participant will receive seven once-weekly telephone calls. All calls will be delivered using a motivational

interviewing approach[35] by a physiotherapist experienced in pulmonary rehabilitation and who has undertaken undertake formal training in motivational interviewing for healthcare. Additional trial-based training and regular fidelity checks using the Motivational Interviewing Treatment Integrity Scale V.4.2.1[36] will also take place. During the telephone calls the physiotherapist will review the home exercise diary; progress the exercise prescription; review symptoms and facilitate self-management of exacerbations; and deliver self-management education via scripted telephonic modules. These structured telephone modules will be used to explore and build motivation for exercise participation.[22 23] The home exercise diary provides participants with a menu of topics related to self-care. Participants will be encouraged to select a topic they feel is relevant to them for discussion with the physiotherapist in their weekly call, providing opportunities for self-management education and goal setting. All participants will have a discussion on managing an acute exacerbation of their COPD and long-term exercise planning, consistent with the goals of centre-based pulmonary rehabilitation programmes. Participants will have access to the 'Better Living with COPD: A Patient Guide' and the C.O.P.E. programme (COPD Online Patient Education) to support topic discussions.

### Data collection and follow-up

Each participant will complete 8 weeks of rehabilitation and a follow-up of 12 months. Outcomes will be assessed at baseline, end of rehabilitation (after 8 weeks) and 12 months following the end of rehabilitation. We will undertake assessments with all participants (unless not possible, eg, due to death, loss-to follow-up, study withdrawal) at 8 weeks and 12 months irrespective of whether a participant has completed their pulmonary rehabilitation programme. Participants will record any healthcare utilisation during the 12 months follow-up (visits to health professionals, hospitalisations) in a diary. Each participant will be telephoned monthly during the 12 months by site research staff to encourage diary completion and capture patient-reported information on healthcare utilisation. These calls represent an additional contact with patients that would not be provided as part of routine clinical practice at trial sites, but the purpose of the call is only for research staff to support collection of data on healthcare use. Hospitalisation and use of other hospital services will be confirmed by medical record audit at 12 months, while 12-month medical and pharmaceutical resource use will be obtained from Services Australia using Pharmaceutical Benefits Scheme (PBS) and Medicare Benefits Schedule (MBS) data. Consent will be sought to use participants' PBS and MBS data at the time of consent to the study. The schedule for assessments and follow-up is provided in table 1.

### Outcomes

#### Primary outcome

*All-cause, non-elective hospitalisation over 12 months:* The number of participants admitted to hospital as an inpatient at least once in the 12 months following pulmonary rehabilitation will be compared between groups. Avoidance of hospitalisation is the research outcome that people with COPD value most highly and is of critical importance to the health system.[37] We have chosen all cause, non-elective hospitalisation due to the challenges in assessing whether an admission is COPD-related in a group where multimorbidity is ubiquitous (median 4 coexisting conditions in our previous trial[23] and the potential for bias in adjudication of respiratory-related admissions significant. We have developed an effective method to collect these data in our previous trial[23] combining medical record audit with monthly telephone calls to participants, to ensure comprehensive coverage of healthcare utilisation.

#### Secondary outcomes

*Programme completion:* The number of participants who complete their allocated pulmonary rehabilitation programme will be compared between groups at 8 weeks.

| Table 1 Schedule of trial follow-up and procedures | | | | |
|---|---|---|---|---|
| Assessment/procedure | Baseline | End rehabilitation programme | Monthly phone calls | 12 months post rehabilitation programme |
| Informed consent | X | | | |
| Demographic information | X | | | |
| Programme completion | | X | | |
| 6-minute walk test | X | X | | X |
| Dyspnoea-12 | X | X | | X |
| Health related quality of life (EQ-5D-5L and CRQ) | X | X | | X |
| Physical activity participation | X | X | | X |
| Healthcare utilisation | | | X | X |
| Economic evaluation | | | X | X |

CRQ, chronic respiratory questionnaire.

Completion is defined as undertaking 70% of planned sessions in accordance with recent recommendations.

*6-minute walk distance (6MWD):* The 6MWD is a validated measure of functional exercise capacity for COPD.[30] The 6MWD predicts both future hospitalisation[38] and survival.[39] It is responsive to change following pulmonary rehabilitation[10] and is an outcome that matters to patients.[40] The test will be performed in accordance with international standards,[30] including two tests at each time point, with the better (ie, longest distance) 6MWD recorded.

*Chronic Respiratory Questionnaire (CRQ):* The CRQ is a disease-specific health-related quality of life measure with domains of dyspnoea, fatigue, mastery and emotional function. It is a patient-centred outcome that is responsive to change following pulmonary rehabilitation.[10] The self-reported version of the CRQ[41] will be used.

*EQ-5D-5L:* This validated generic quality of life measure[42] is used to estimate health benefits in terms of quality-adjusted life-years (QALYs),[43] and is recommended for economic analyses.

*Dyspnoea-12:* This a global measure of breathlessness severity that captures both the physical and affective components of dyspnoea. It has excellent internal consistency, validity and reliability in COPD.[44]

*Physical activity participation:* This is an important measure of behaviour change following pulmonary rehabilitation. We will measure time spent in moderate to vigorous physical activity and sedentary time, using the ActiGraph GT3X, a waist-worn, tri-axial accelerometer that has been validated in people with COPD.[45] Seven days of monitoring (in the week following the 8-week or 12-month assessment) will be performed to allow for at least four valid days of at least 8 hours wear time.[46]

*Healthcare utilisation across 12 months:* Healthcare utilisation will be recorded monthly in participant diaries and telephone record sheets kept by site research staff. We will confirm healthcare utilisation from hospital records, MBS and PBS data.

## Sample size

The primary outcome is unplanned hospitalisation (inpatient admissions) in the 12 months following pulmonary rehabilitation. In our published Phase II trial,[23] 57% of the centre-based group were free from hospitalisation at 12 months, whereas 78% of rehabilitation completers were free from hospitalisation (regardless of group). Completion was 91% for HomeBase, and 49% for centre based.[23] The novel aspect of the current study is that intervention participants will be offered a choice of models. Based on our experience recruiting for Home-Base, we expect that 75% of individuals may select a home programme of whom 91% will complete, and 25% will select centre-based of whom 49% will complete. We thus anticipate a completion rate of 80% in the intervention (choice) group.

At 80% completion, our data indicate that 72% of the intervention group would remain free from admission at 12 months, compared with the observed rate of 57% in centre-based (control) participants.[23] We thus expect 15% reduction in hospitalisations in the intervention group (choice of home-based or centre-based rehabilitation) compared with control (centre-based rehabilitation only). To detect a difference in remaining free from hospitalisation between intervention and control groups of 15% (72% vs 57%) with 80% power and a two-sided 0.05 significance level, 312 participants (156 in each group) are required. Adjusting for clustering by programme (intracluster correlation coefficient of 0.01, based on our published two-site trial and 35 participants per cluster), the total required sample is 418. Our previous study had 10% loss to follow-up, so to allow for this we will recruit a total of 490 participants from 14 pulmonary rehabilitation programmes.

## Statistical analysis

Generalised linear mixed models will be used to analyse the primary outcome of hospitalisation, and linear mixed models will be used for continuous outcomes. These will account for correlation between participants within a cluster. Analyses will be based on intention-to-treat and the level of significance will be set at p<0.05.

## Economic evaluation

Economic evaluation will follow the protocol developed for our Phase II trial.[22 23]

*Cost comparisons:* Cost comparisons will include comparison of per person costs, including direct (health system) and indirect (personal) healthcare costs for the two groups. Direct costs will include staff time, consumables, communications and overheads. Intervention costs will include staff inputs by duration, type and resource use (including troubleshooting and support) and equipment. Personal costs will include transportation, travel time and impact of the intervention on the economic activities of other household members. Health system costs will include visits to the general practitioner, specialist or emergency department; use of chronic disease services; and hospitalisation.

*Cost-effectiveness analysis:* An incremental cost-effectiveness analysis will be undertaken to compare differences in costs between the two groups with differences in QALYs and number of hospitalisations in the 12-month follow-up period. For the former, a utility index will be calculated from the EQ-5D-5L by applying a 'social tariff', which then enables estimation of health benefits in terms of QALYs. The conversion to QALYs will be based on the assumption that for health states measured at successive points of time, the duration of each health state is exactly one half of the time interval between the two measurements.[47] For the latter, the indicator of cost-effectiveness can be interpreted as the incremental cost of avoiding a hospitalisation.

## Process evaluation

A process evaluation, informed by the Medical Research Council Guidance on Process Evaluations of complex

**Table 2** Evaluation of implementation using the RE-AIM (Reach, Effectiveness, Adoption, Implementation and Maintenance) framework

| RE-AIM element | Quantitative and qualitative data |
|---|---|
| **Reach** | Intervention participation rates (number who chose home-based and centre-based rehabilitation). |
| **Effectiveness** | Programme completion (>70% sessions attended), clinical outcomes. |
| **Adoption** (further explored using the TDF) | Barriers and facilitators to implementation. |
| **Implementation** | Programme components delivered (eg, exercise training/progression), use of exercise diaries. |
| **Maintenance** (further explored using the TDF) | Intent to continue offering programme choice, modifications made. |

TDF, Theoretical Domains Framework.

interventions,[48] is also planned for this trial. Offering a choice of programme models represents a departure from traditional care. Thus, understanding both clinician and patient experiences of this new approach is critical to guide scale up and future implementation. A detailed protocol for the process evaluation will be published separately. Briefly, the process evaluation will undertake a theory-based approach using the Theoretical Domains Framework (TDF).[49] The deductive coding using the TDF will allow the team to understand clinician or patient behaviours that act as barriers or facilitators to implementation of HomeBase, and to develop a compendium of implementation strategies for future scale up. Implementation will be further explored using the RE-AIM (Reach, Effectiveness, Adoption, Implementation and Maintenance) framework[50] with collection of both quantitative and qualitative data throughout the trial (table 2) to provide insights into translation and behaviour change success in intervention clusters.

## ETHICS AND DISSEMINATION

This protocol (V.2, 16 August 2020) has received approval from Alfred Hospital Human Research Ethics Committee (379/19). Local research ethics committee approvals have also been obtained at sites where the National Mutual Acceptance scheme does not apply. Each site is also required to obtain local governance approval prior to commencement of data collection. Site investigators will be informed by the trial coordinator of any necessary modifications to the protocol and amendments to local ethics and/or governance approval. Potential participants at each site may be in a dependent relationship with pulmonary rehabilitation staff. For this reason pulmonary rehabilitation staff will only provide initial information about the trial to a potential participant, either at the time of initial assessment or over the telephone. If the potential participant is interested in receiving further detailed information, or providing informed consent, a researcher who is not in a dependent relationship with the potential participant will undertake this. The trial findings will be published in peer-reviewed journals and submitted to national and international conferences for presentation. The authorship of any publications will adhere to the guidelines established by the International Committee of Medical Journal Editors. We will disseminate results to patients with COPD across Australia through Lung Foundation Australia's e-newsletter and website, and to health professionals through Thoracic Society of Australia and New Zealand's e-news, scientific conferences and webinar programme.

## Safety monitoring and reporting

All sites will maintain an adverse event log for the trial period. An adverse event in the trial is being defined as any unfavourable, unintended diagnosis, symptom, sign (including an abnormal laboratory finding), syndrome or disease that occurs during the trial, having been absent at baseline, or if present at baseline, appears to have worsened. A serious adverse event is defined as any adverse event leading to (i) death; (ii) serious deterioration in health resulting in a life-threatening illness or injury, or permanent impairment of a body structure or a body function; (iii) hospitalisation or prolongation of existing hospitalisation; (iv) medical or surgical intervention to prevent life-threatening illness; (v) fetal distress, fetal death or a congenital abnormality. Sites will inform the trial coordinator of any adverse event that necessitates a change to the protocol or participant information and consent form, is unexpected, or any serious adverse events. The trial coordinator will ensure the data and safety monitoring board (DSMB), relevant research ethics committees or governance offices are informed of any serious adverse events where necessary. The DSMB, consisting of a respiratory physician, two physiotherapists and a statistician, will make the final decision on the relatedness of any adverse events to the trial interventions or procedures. Unless otherwise determined by trial events, the DSMB will convene biannually.

## Data management

Hard copy original data collection forms will be stored in a locked filing cabinet within a locked office at each trial site. Electronic data will be stored centrally in a

purpose-built on-line database (www.adeptrs.com), with encryption and password protection. No identifying information will be stored in the online database. Electronic data for all sites will be accessible by the coordinating principal investigator and the trial coordinator. Site specific investigators will only have access to data relating to their individual site.

## Patient and public involvement

The experiences of HomeBase participants in our previous Phase II trial[24] were critical to formulating the current proposal. Hospitalisation has been chosen as the primary outcome of this trial because it is the research outcome that is most meaningful for patients with COPD.[37] Two patients with COPD will sit on the trial steering committee. All patient-facing trial resources (eg, flyers, Participant Information and Consent Forms, diaries) have undergone review by patients with COPD via Lung Foundation Australia's COPD Patient Advisory Group.

## Trial status

Trial recruitment began in March 2021 and is ongoing.

## DISCUSSION

This research will be the first randomised controlled trial to determine the clinical benefits and healthcare costs of offering patients a choice of two pulmonary rehabilitation locations (home-based or centre-based) compared with offering only traditional centre-based models. Until recently, there has been very little 'choice' for patients referred to pulmonary rehabilitation; the delivery model of centre-based pulmonary rehabilitation has largely been a 'one size fits all' that is inconsistent with the personalised approach of modern medicine.

The adoption of alternative models for pulmonary rehabilitation will rely on demonstration of comparable or better clinical outcomes compared with those of traditional pulmonary rehabilitation programmes, which can be delivered in a cost-effective manner.[21] A novel aspect of this trial is that our intervention will offer patients with COPD the choice of two rehabilitation models that are underpinned by strong efficacy data, an approach that provides patients with increased opportunities to attain the benefits of pulmonary rehabilitation. Our previous economic analysis[51] suggested that completion of pulmonary rehabilitation is an independent predictor of lower healthcare costs in COPD.

Based on head-to-head trial evidence of home versus outpatient-based pulmonary rehabilitation, the 2017 Australian Pulmonary Rehabilitation guidelines recommended home programmes as an alternative to hospital-based programmes.[28] The offer of a home-based programme has long been a limited option in pulmonary rehabilitation services.[16] The COVID-19 pandemic had led to many pulmonary rehabilitation services rapidly transitioning to home-based models but there remains little information about how best to implement these models in a clinical setting. Home-based programmes have the potential to address many of the patient-related and system-related barriers to attending traditional centre-based pulmonary rehabilitation models in COPD including improvements to access (eg, greater scope in programme delivery) and uptake (allowing patient preference for home-based care, reducing barriers related to travel and disability). Targeted implementation is required to translate the clinical efficacy of home-based programmes to important health outcomes in routine practice. Implementation of home-based programmes alongside the traditional outpatient centre-based models may allow evidence-based pulmonary rehabilitation to be more accessible to patients. A process evaluation will be conducted alongside the cluster randomised controlled trial in order to understand barriers and facilitators to offering choice of pulmonary rehabilitation models at trial sites, and to provide data that will guide future implementation.

This trial will provide the evidence needed to underpin future guidelines and policy decisions for pulmonary rehabilitation. If successful, the findings will inform a key change in the future of pulmonary rehabilitation services worldwide, where delivery includes more choices for patients and opportunities for greater personalisation of programmes.

**Author affiliations**
[1]Respiratory Research@Alfred, Central Clinical School, Monash University, Melbourne, Victoria, Australia
[2]Department of Physiotherapy, Alfred Health, Melbourne, Victoria, Australia
[3]Institute for Breathing and Sleep, Heidelberg, Victoria, Australia
[4]The Nossal Global Institute for Global Health, University of Melbourne, Melbourne, Victoria, Australia
[5]Department of Clinical Neuroscience, Central Clinical School, Monash University, Melbourne, Victoria, Australia
[6]Allied Health (Occupational Therapy), Alfred Health, Melbourne, Victoria, Australia
[7]Statistical Consulting Centre, The University of Melbourne, Melbourne, Victoria, Australia
[8]Department of Public Health, La Trobe University, Bundoora, Victoria, Australia
[9]Department of Respiratory and Sleep Medicine, Austin Health, Heidelberg, Victoria, Australia
[10]Department of Medicine, University of Melbourne, Melbourne, Victoria, Australia

**Contributors** AEH, CFM, AM, NAL and NC conceived the study; AEH, AWJ, AM, NAL, NC, GH, PO and CFM developed the protocol and methods; AEH, AWJ, AM, NAL, NC, GH, PO and CFM prepared the manuscript and approved the final submitted version.

**Funding** This work is supported by the Commonwealth of Australia Medical Research Future Fund grant MRF1176491. This funding source had no role in the design of this study and will not have any role during its execution, analyses, interpretation of the data or decision to submit results.

**Competing interests** AEH, AM, CFM, GH, NAL, NC and PO declare grant funding from the Commonwealth of Australia Medical Research Future Fund paid to their institution for the conduct of the trial. CFM declares non-financial board roles as Chair COPD National Programme, Lung Foundation Australia and Medical Director Institute for Breathing and Sleep. No other competing interests are declared.

**Patient and public involvement** Patients and/or the public were involved in the design, or conduct, or reporting, or dissemination plans of this research. Refer to the Methods section for further details.

**Patient consent for publication** Not applicable.

**Provenance and peer review** Not commissioned; externally peer reviewed.

**Open access** This is an open access article distributed in accordance with the Creative Commons Attribution 4.0 Unported (CC BY 4.0) license, which permits others to copy, redistribute, remix, transform and build upon this work for any purpose, provided the original work is properly cited, a link to the licence is given, and indication of whether changes were made. See: https://creativecommons.org/licenses/by/4.0/.

**ORCID iDs**

Anne E Holland http://orcid.org/0000-0003-2061-845X
Arwel W Jones http://orcid.org/0000-0003-1689-8065
Natasha A Lannin http://orcid.org/0000-0002-2066-8345
Narelle Cox http://orcid.org/0000-0002-6977-1028
Christine F McDonald http://orcid.org/0000-0001-6481-3391

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
