## [Reviewer comments · BMJ Open]

ARTICLE DETAILS

TITLE (PROVISIONAL)	Implementing a choice of pulmonary rehabilitation models in chronic obstructive pulmonary disease (HomeBase2 trial): protocol for a cluster randomised controlled trial
AUTHORS	Holland, Anne E.; Jones, Arwel; Mahal, Ajay; Lannin, Natasha; Cox, Narelle; Hepworth, Graham; O'Halloran, Paul; McDonald, Christine

VERSION 1 – REVIEW

REVIEWER	Martin Heine Stellenbosch University, Institute of Sports and Exercise Medicine
REVIEW RETURNED	17-Nov-2021

GENERAL COMMENTS	Thank you for the opportunity to review your submission titled “Implementing a choice of pulmonary rehabilitation models in COPD (HomeBase2 trial): protocol for a cluster randomised controlled trial.” To my understanding, the aim of this implementation trial is to determine the clinical benefits and health care cost associated with providing study participants centre-based rehabilitation or the option of centre-based rehabilitation versus a home-based rehabilitation program. The protocol is well written, and rooted in a series of research activities leading up to this study, in conjunction with existing evidence with respect to the efficacy of pulmonary rehabilitation in this patient population. That being said, my main concern is a incongruency between the aim (i.e. implementation) relative to the main focus of the protocol (i.e. effectiveness. Effectiveness is just one component in the RE-AIM framework, and it would strengthen the protocol if the other components would take a stronger centre stage. Moreover, we know that pulmonary rehabilitation is effective (home-based or centre-based), hence methodologically, I would have expected a strong focus on the completion rate (primary outcome of interest), in conjunction with the economic analyses and process evaluation (in line with the argument provided in 148 – 152). In addition to the above, some minor concerns or points for consideration are listed below: - Point 4 in the “strengths and limitations” overview is not entirely clear.- While I understand that recruitment has already started, I was wondering whether the authors have considered a stepped-wedge trial design rather. Such a design would have had the added benefit that upon study completion, all sites would have implemented the choice option and minimizing the need for subsequent implementation initiatives at those sites. Furthermore, a stepped-wedge design allows for a better evaluation of time (due to longer
---

	follow-up periods for those crossing over early in the trial) as a factor in the long-term efficacy of pulmonary rehabilitation (or implementation of the choice model), and would also have minimized potential other site-specific confounders (e.g., rurality) currently not accounted for?  - The notion that patients with comorbidities which preclude exercise training are excluded should be made more specific, given a) the high multimorbidity in this patient population, and b) patients with some comorbidities could be considered for centre-based PR, while excluded for home-based PR. - Please provide a definition for “hospitalisation” - Safety and monitoring. I guess your primary outcome (non-elective hospitalisation) is also a serious adverse event requiring reporting to the ethics committee. How would this practically work? - There is no clear rationale provided to determine strata based on hospital, non-hospital, while there are other cluster-level features that may have an important impact, for example rurality of the setting, prevalent demographic in the region, amongst others. Please clarify. All the best with this important and huge undertaking.
--	---

REVIEWER	Jean-Marie Grosbois FormAction Santé
REVIEW RETURNED	06-Dec-2021

GENERAL COMMENTS	Anne Holland and her team have submitted an ambitious protocol for a multicentric cluster randomized, controlled, assessor-blinded trial, that will be conducted in a large sample size of patients with COPD (n=490). As usual with Holland and her team, the protocol is well written, rich with a lot of key references from the same team, showing their expertise in home-based pulmonary rehabilitation setting. The aim of the RCT is to compare PR effectiveness (on hospitalisation: first outcome, exercise tolerance, health related quality of life, dyspnea and physical activity) and attendance/completion for programs implementing a choice of location (home or centre) to those offering centre-based only. Although the short-term effectiveness of PR or home-based PR is no longer a question in patients with COPD, the real originality of the present protocol is to let the patient choosing between a home-based PR or a center-based PR. As mentioned by the authors, offering alternative models to traditional intervention might increase patients access to PR and reduce health care costs. I have no concern about the design of the study, the ethics approvals of statistical analysis. Please find below few suggestions/questions that should be bring to the authors' attention. 1/ I have difficulty calling 'home-based PR' an intervention that has only one initial supervised home visit following by 7-week phone calls. Could 'telerehabilitation' be a better description? In their previous studies Holland and her team used both home-based and telerehabilitation for describing the same intervention. What's their position? 2/ If I understand well, 7 of the 14 centre-based PR services will also provide home-based PR. Did these centers already provided home-based PR before the study? If no, how were they trained? For how long? And what was the cost? I raise these questions because I disagree with the authors statement page 23, lines 549-550: 'the
--

findings of this trial can be readily implemented into clinical practice worldwide'. Unfortunately, home-based PR is not well developed worldwide and usually center-based PR cannot provide a choice of location to patients with COPD. Home-based PR or telerehabilitation has still a cost that needs to be recognize worldwide.

3/ I would like to congrats the authors about the colossal work they are going to do, especially with the first outcome. In my opinion, looking at the all-cause, non-elective hospitalisations in the 12 months following PR is relevant and will require rigor for both the patients and the research team (but certainly with a high risk of non-exhaustiveness). It would have been interesting to also analyse the hospitalisations during the year before the intervention. Will you be able to obtain this information?

In my opinion you should mention that the 12-month phone calls follow-up after PR is not traditional with PR or home-based PR. Few centers can afford a year of monthly phone calls follow-up.

4/ The authors should detail the home-based physical activity training using the pedometer as a feedback (page 12, lines 272-275. I understood that the walking speed will be set according to the 6MWT but what about to number of steps that participants will have to do? Is the pedometer will be the same in all 14 centers? Will participants use the pedometer during the full year after PR?

5/ Page 16 line 368 authors mentioned that daily physical activity will be measured with an Actigraph GT3X during 7 days. Although this device has been validated in COPD, actigraphy is not well sensitive to PR, especially at 12-months follow-up.

Lines 369-370: 'We will measure time spent in moderate to vigorous physical activity and sedentary time'. In my experience, patients with COPD spend no time in vigorous physical activity intensity and very few minutes a day in moderate intensity. Light physical activity intensity should be looked at as well. What about the range of wearing time for valid measurement?

Table 1. Physical activity will be measured at baseline, at the end of PR and 12 months after PR. At the end of PR will it be the last week of PR, or the week after PR?

6/Page 11, lines 241-244 "At the completion of the rehabilitation period participants will be offered the opportunity to join a supervised exercise maintenance program to promote ongoing exercise adherence in accordance with existing clinical practice at the trial site". I am concern about this detail. Is all participants (control group and intervention group) will have the opportunity to join a supervised exercise maintenance program after PR? Will you compare participants who enrolled in maintenance program to participants who will not? The maintenance program will definitely impact on your first outcome (hospital admission in the 12 months following PR). Precisions should be added.

7/ Page 14, lines 318-320 "Assessments will be undertaken at 8 weeks and 12 months irrespective of whether a participant has completed their pulmonary rehabilitation program". I do not understand this statement. What about drop-out during PR for death? What about participants that drop-out during PR and do not want to see the PR team again? I suggest you rephrase this sentence.

Minor comment

	10/ Page 21, line 508 and page 19 line 448: I am surprised to review a protocol that has already been in progress for over a year.
--	--

VERSION 1 – AUTHOR RESPONSE

Reviewer comments	Response
Thank you for the opportunity to review your submission titled “Implementing a choice of pulmonary rehabilitation models in COPD (HomeBase2 trial): protocol for a cluster randomised controlled trial.” To my understanding, the aim of this implementation trial is to determine the clinical benefits and health care cost associated with providing study participants centre-based rehabilitation or the option of centre-based rehabilitation versus a home-based rehabilitation program. The protocol is well written, and rooted in a series of research activities leading up to this study, in conjunction with existing evidence with respect to the efficacy of pulmonary rehabilitation in this patient population. That being said, my main concern is a incongruency between the aim (i.e. implementation) relative to the main focus of the protocol (i.e. effectiveness. Effectiveness is just one component in the RE-AIM framework, and it would strengthen the protocol if the other components would take a stronger centre stage. Moreover, we know that pulmonary rehabilitation is effective (home-based or centre-based), hence methodologically, I would have expected a strong focus on the completion rate (primary outcome of interest), in conjunction with the economic analyses and process evaluation (in line with the argument provided in 148 – 152).	Thank you very much for your comments. Thank you for raising this. First, we wanted to emphasise that our intention is to publish a separate protocol on the process evaluation (see line 456-467). However, we appreciate though that our aims at the end of introduction drew too much attention to clinical effectiveness. We have addressed this section (line 158-168) so that the reader understands that this a hybrid effectiveness-implementation trial whereby assessment of clinical effectiveness is only one objective of the evaluation. We have also added Table 2 the manuscript, which provides the reader with quantitative and qualitative data from the trial that will be evaluated according to the RE-AIM framework.
Point 4 in the “strengths and limitations” overview is not entirely clear.	This point has been clarified. See line 84-88.
While I understand that recruitment has already started, I was wondering whether the authors have considered a stepped-wedge trial design rather. Such a design would have had the added benefit that upon study completion, all sites would	We agree that there are attractive features of a stepped-wedged trial design. However, we were concerned that implementing a choice of home-based pulmonary rehabilitation is not like other interventions, as it requires training including

have implemented the choice option and minimizing the need for subsequent implementation initiatives at those sites. Furthermore, a stepped-wedge design allows for a better evaluation of time (due to longer follow-up periods for those crossing over early in the trial) as a factor in the long-term efficacy of pulmonary rehabilitation (or implementation of the choice model), and would also have minimized potential other site-specific confounders (e.g., rurality) currently not accounted for?	motivational interviewing training for some of the local staff (if they so choose to support call delivery - also see response to reviewer 2 below), service re-organisation and a significant shift in attitude (more around behaviour change than delivery of a program per se). Hence, we felt if the training started during the control period it may have affected the delivery of usual care.
The notion that patients with comorbidities which preclude exercise training are excluded should be made more specific, given a) the high multimorbidity in this patient population, and b) patients with some comorbidities could be considered for centre-based PR, while excluded for home-based PR.	We used this criterion in our previous trial whereby this represented only 3% of those assessed for eligibility (Holland et al. Home-based rehabilitation for COPD using minimal resources: a randomised, controlled equivalence trial. Thorax. 2017;72(1):57-65) There will be no difference across home-based PR or centre-based PR with regards to eligibility assessment for the study. The offer of choice in the intervention sites occurs after participants consent for the trial (i.e. following eligibility assessment). All participants must attend a face-to-face assessment at baseline. This criterion is in line with standard delivery of pulmonary rehabilitation programs. For both home and centre PR it is about whether it is safe to undertake exercise testing (6MWT) for prescription of exercise. There is now reference to the following citation in the manuscript (line 226-227), which refers to absolute and relative contraindications for the 6MWT (Holland et al. An official European Respiratory Society/American Thoracic Society technical standard: field walking tests in chronic respiratory disease. Eur Respir J. 2014;44(6):1428-46.)
Please provide a definition for "hospitalisation"	Hospitalisation in this study is defined as an inpatient admission. Please see addition to lines 352 and 398.
Safety and monitoring. I guess your primary outcome (non-elective hospitalisation) is also a serious adverse event requiring reporting to the ethics committee. How would this practically	Individual reports of serious adverse events are only required to be reported to the lead ethics committee if they are related to the intervention (i.e. delivery of pulmonary rehabilitation) or

work?	research procedures. Trial sponsor will hold on to this information if deemed unrelated. If a serious adverse event that is related to interventions or research procedures does occur correspondence with the ethics committee will be undertaken by the trial coordinator and other members of the team involved in data analysis will remain blinded to cluster allocations.
There is no clear rationale provided to determine strata based on hospital, non-hospital, while there are other cluster-level features that may have an important impact, for example rurality of the setting, prevalent demographic in the region, amongst others. Please clarify.	We have added our rationale in lines 191-198. Given the number of clusters we wanted to limit the number of stratification factors and prioritise the most important. We wanted to prioritise location/type of pulmonary rehabilitation program. This was due to two main reasons:  1) help control for readiness for implementation of choice of models whereby local staff capacity and infrastructure may vary across a hospital or non-hospital site 2) help control for well-known barriers for patients in attending centre-based pulmonary rehabilitation including travel/distance/location and (non-hospital local community programs may be situated closer to homes of patients, accessibility of car parking at hospital) and reasons for why patients with COPD may opt for home-based pulmonary rehabilitation. See:  Cecins et al Community-based pulmonary rehabilitation in a non-healthcare facility is feasible and effective. Chron Respir Dis. 2017;14(1):3-10; Keating et al What prevents people with chronic obstructive pulmonary disease from attending pulmonary rehabilitation? A systematic review. Chron Respir Dis. 2011;8(2):89-99; Lahham et al. Home-based pulmonary rehabilitation for people with COPD: A qualitative study reporting the patient perspective. Chron Respir Dis. 2018;15(2):123-130 The large and medium sized hospitals that deliver pulmonary rehabilitation programs in Australia are situated in urban/metropolitan areas. Hence, with our local context in mind, we feel the that our type of centre-based program strata will also help to balance sociodemographic profiles between groups and the potential effects that such factors

	may have on trial outcomes (e.g. Terry et al. Factors contributing to COPD hospitalisations from 2010 to 2015: Variation among rural and metropolitan Australians. Clin Respir J. 2019;13(5):306-313).
All the best with this important and huge undertaking.	Thank you for your well wishes. We appreciate your time in conducting your review of our protocol.
Reviewer 2	
Anne Holland and her team have submitted an ambitious protocol for a multicentric cluster randomized, controlled, assessor-blinded trial, that will be conducted in a large sample size of patients with COPD (n=490). As usual with Holland and her team, the protocol is well written, rich with a lot of key references from the same team, showing their expertise in home-based pulmonary rehabilitation setting. The aim of the RCT is to compare PR effectiveness (on hospitalisation: first outcome, exercise tolerance, health related quality of life, dyspnea and physical activity) and attendance/completion for programs implementing a choice of location (home or centre) to those offering centre-based only. Although the short-term effectiveness of PR or home-based PR is no longer a question in patients with COPD, the real originality of the present protocol is to let the patient choosing between a home-based PR or a center-based PR. As mentioned by the authors, offering alternative models to traditional intervention might increase patients access to PR and reduce health care costs. I have no concern about the design of the study, the ethics approvals of statistical analysis. Please find below few suggestions/questions that should be bring to the authors' attention.	Thank you very much for your comments.
I have difficulty calling 'home-based PR' an intervention that has only one initial supervised home visit following by 7-week phone calls. Could 'telerehabilitation' be a better description? In their previous studies Holland and her team used both home-based and telerehabilitation for describing the same intervention. What's their position?	The current protocol utilises our home-based pulmonary rehabilitation program, which is a simple low-cost model delivered primarily with telephone support. Our Phase II randomised controlled trial (Holland et al. Home-based rehabilitation for COPD using minimal resources: a randomised, controlled equivalence trial. Thorax. 2017;72(1):57-65) showed that this home-based pulmonary rehabilitation model delivers equivalent clinical outcomes to traditional

	centre-based rehabilitation We developed this home-based model to improve access to pulmonary rehabilitation. You are correct in that we have also developed a telerehabilitation model (Cox NS et al. Telerehabilitation for chronic respiratory disease: a randomised controlled equivalence trial. Thorax. 2021:thoraxjnl-2021-216934). However, that model is distinct to the home-based program. The telerehabilitation model is more complex, as it utilises internet technology to provide real time supervision to virtual groups of up to six participants. Given the differences between these models, particularly in terms of complexity, we have chosen to retain the term 'home-based PR' for the current trial.
If I understand well, 7 of the 14 centre-based PR services will also provide home-based PR. Did these centers already provided home-based PR before the study? If no, how were they trained? For how long? And what was the cost? I raise these questions because I disagree with the authors statement page 23, lines 549-550: 'the findings of this trial can be readily implemented into clinical practice worldwide'. Unfortunately, home-based PR is not well developed worldwide and usually center-based PR cannot provide a choice of location to patients with COPD. Home-based PR or telerehabilitation has still a cost that needs to be recognize worldwide	No, the centres are not already providing home-based pulmonary rehabilitation. The 7-week telephone calls are delivered by a central team consisting of physiotherapists trained in motivational interviewing via an online course over several weeks or an in person/teleconference event over 2-3 full days followed by 1:1 coaching sessions delivered by a qualified MINT (Motivational Interviewing Network of Trainers) trainer to develop and confirm proficiency. This central team may include staff from the 7 intervention sites if they choose to do so. All home visits will be undertaken by local staff from the 7 intervention sites. Specific training for this home visit takes less than half a day where staff have existing pulmonary rehabilitation experience. We would like to emphasise that whilst this additional training is required, delivery of this program is within the scope of practice of health professionals employed in pulmonary rehabilitation, however some service re-organisation is required. Our trial will examine whether a choice of pulmonary rehabilitation models is effective/cost-effective and look to identify the barriers or facilitators to service delivery. We have deleted the sentence 'the findings of this trial can be readily implemented into clinical practice' and feel there is no need to add a replacement sentence as the text reads well with

	the preceding and subsequent sentence. See line 569-573.
I would like to congrats the authors about the colossal work they are going to do, especially with the first outcome. In my opinion, looking at the all-cause, non-elective hospitalisations in the 12 months following PR is relevant and will require rigor for both the patients and the research team (but certainly with a high risk of non-exhaustiveness). It would have been interesting to also analyse the hospitalisations during the year before the intervention. Will you be able to obtain this information? In my opinion you should mention that the 12-month phone calls follow-up after PR is not traditional with PR or home-based PR. Few centers can afford a year of monthly phone calls follow-up.	Yes, information on hospitalisations in the 12 months before the intervention will be available. Sites are funded by the Sponsor to deliver these monthly follow-up calls. The phone calls are about collecting data for the health economic analysis, not clinical care, but we acknowledge it is an additional contact that would not be provided in clinical practice. This has been revised, see lines 339-342.
The authors should detail the home-based physical activity training using the pedometer as a feedback (page 12, lines 272-275. I understood that the walking speed will be set according to the 6MWT but what about to number of steps that participants will have to do? Is the pedometer will be the same in all 14 centers? Will participants use the pedometer during the full year after PR?	We have clarified the text here. See lines 290-292. Patients may choose their distance to be guided by km or steps on the pedometer, but key is to use the distance to inform speed (i.e. amount of km or steps covered in a time period) during the home-based program. Only participants in the 7 intervention centres who choose home-based pulmonary rehabilitation will receive a pedometer. Pedometers will not be returned at the end of program, participants will have access to them for the entire duration of the study.
Page 16 line 368 authors mentioned that daily physical activity will be measured with an Actigraph GT3X during 7 days. Although this device has been validated in COPD, actigraphy is not well sensitive to PR, especially at 12-months follow-up. Lines 369-370: 'We will measure time spent in moderate to vigorous physical activity and sedentary time'. In my experience, patients with COPD spend no time in vigorous physical activity intensity and very few minutes a day in moderate intensity. Light physical activity intensity should be looked at as well. What about the range of wearing time for valid measurement? Table 1. Physical activity will be measured at baseline, at the end of PR and 12 months after PR. At the end of PR will it be the last week of PR, or the week after PR?	For the end-rehabilitation timepoint, physical activity will be monitored in the week following the program. Valid wear time will be in line with the latest recommendations for COPD populations. We clarified this in lines 390-392. Our choice of time in MVPA and sedentary follows on from our previous Phase II trial (Holland et al. Home-based rehabilitation for COPD using minimal resources: a randomised, controlled equivalence trial. Thorax. 2017;72(1):57-65). This is supported by studies showing that patients classified as physically active (i.e., those who reach MVPA recommendations) in combination with a non-sedentary lifestyle are those that present with markedly better clinical conditions (e.g. Schneider et al. Sedentary Behaviour and Physical Inactivity

	in Patients with Chronic Obstructive Pulmonary Disease: Two Sides of the Same Coin? COPD. 2018;15(5):432-438). Time in MVPA has long been advocated as the intensity which brings about the “health enhancing’ effects of physical activity (as per global physical activity guidelines). More recently, reduction in sedentary behaviour has been shown to produce health benefits even if changes in MVPA are not achieved. See: Biswas et al. Sedentary time and its association with risk for disease incidence, mortality, and hospitalization in adults: a systematic review and meta-analysis. Ann. Intern. Med. 2015; 162: 123–132. Chau JY et al. Daily sitting time and all-cause mortality: a meta-analysis. PLoS One 2013; 8: e80000. van der Ploeg et al. Sitting time and all-cause mortality risk in 222 497 Australian adults. Arch. Intern. Med. 2012; 172: 494–500. We take your point that time in light PA could be reported but any increase in light PA is likely to be observed with a reduction in sedentary behaviour.
Page 11, lines 241-244 “At the completion of the rehabilitation period participants will be offered the opportunity to join a supervised exercise maintenance program to promote ongoing exercise adherence in accordance with existing clinical practice at the trial site”. I am concern about this detail. Is all participants (control group and intervention group) will have the opportunity to join a supervised exercise maintenance program after PR? Will you compare participants who enrolled in maintenance program to participants who will not? The maintenance program will definitely impact on your first outcome (hospital admission in the 12 months following PR). Precisions should be added.	We have clarified this section of text. See lines 250-253. This is an implementation trial so we do not want to interfere with usual delivery of pulmonary rehabilitation services. Many pulmonary rehabilitation programs in Australia provide an offer of/referral to a supervised maintenance program at the end of pulmonary rehabilitation. We are not convinced that maintenance programs will definitely impact on our primary outcome (all-cause hospitalisation), the evidence in this area remains uncertain. See: Jenkins et al. Efficacy of supervised maintenance exercise following pulmonary rehabilitation on health care use: a systematic review and meta-

	analysis. Int J Chron Obstruct Pulmon Dis. 2018) Malaguti et al. Supervised maintenance programmes following pulmonary rehabilitation compared to usual care for chronic obstructive pulmonary disease. Cochrane Database Syst Rev.17;8(8):CD013569). We also found no evidence that attendance at maintenance made a difference in our Phase II trial (Holland et al. Home-based rehabilitation for COPD using minimal resources: a randomised, controlled equivalence trial. Thorax. 2017;72(1):57-65) comparing home-based pulmonary rehabilitation to centre-based pulmonary rehabilitation. Nevertheless, we will collect data on who goes (or not) to maintenance programs and will consider such factors in sensitivity analysis.
Page 14, lines 318-320 "Assessments will be undertaken at 8 weeks and 12 months irrespective of whether a participant has completed their pulmonary rehabilitation program". I do not understand this statement. What about drop-out during PR for death? What about participants that drop-out during PR and do not want to see the PR team again? I suggest you rephrase this sentence.	We have re-phrased this sentence, see line 331-333. Here we wanted to indicate that participants will be asked to complete all assessments even if they did not complete the pulmonary rehabilitation program, to fulfil the requirements of intention to treat analysis. In previous trials we have found that participants are often willing to return for an assessment for research, even if they were not able to complete pulmonary rehabilitation.
Page 21, line 508 and page 19 line 448: I am surprised to review a protocol that has already been in progress for over a year.	COVID-19 has certainly had an impact on research worldwide. We were pleased to be able to commence the protocol in 2021 due to Australia's COVID elimination strategy keeping community transmission lower than many parts in the world. Nevertheless, lockdowns and restrictions placed on pulmonary rehabilitation programs has meant we are now (like many other protocols) still in the early stages of recruitment for the study.

VERSION 2 – REVIEW

REVIEWER	Martin Heine Stellenbosch University, Institute of Sports and Exercise Medicine
REVIEW RETURNED	18-Feb-2022
GENERAL COMMENTS	Thank you for addressing the comments provider by myself and reviewer 2; All the best with this interesting and huge undertaking.